# Novel Ultrasound-Guided Cervical Intervertebral Disc Injection of Platelet-Rich Plasma for Cervicodiscogenic Pain: A Case Report and Technical Note

**DOI:** 10.3390/healthcare10081427

**Published:** 2022-07-29

**Authors:** King Hei Stanley Lam, Chen-Yu Hung, Tsung-Ju Wu, Wei-Hung Chen, Tony Kwun Tung Ng, Jui-An Lin, Yung-Tsan Wu, Wai Wah Lai

**Affiliations:** 1The Department of Clinical Research, The Hong Kong Institute of Musculoskeletal Medicine, Hong Kong, China; laiwwm@gmail.com; 2The Department of Family Medicine, Faculty of Medicine, The Chinese University of Hong Kong, Hong Kong, China; 3The Department of Family Medicine, LKS Faculty of Medicine, The University of Hong Kong, Hong Kong, China; 4Center for Regional Anesthesia and Pain Medicine, Wan Fang Hospital, Taipei Medical University, Taipei 11031, Taiwan; tonyktng@gmail.com (T.K.T.N.); juian.lin@tmu.edu.tw (J.-A.L.); 5Center for Regional Anesthesia and Pain Medicine, Chung Shan Medical University Hospital, Taichung 40201, Taiwan; 6Department of Physical Medicine and Rehabilitation, National Taiwan University Hospital, Bei-Hu Branch, Taipei 10845, Taiwan; chenyu810@gmail.com; 7Graduate Institute of Basic Medical Science, China Medical University, Taichung 40402, Taiwan; 1520798@cch.org.tw; 8Department of Physical Medicine and Rehabilitation, Changhua Christian Hospital, Changhua City 50006, Taiwan; 9Department of Anesthesiology, E-Da Hospital, Kaohsiung City 82445, Taiwan; gnr.chen@gmail.com; 10Pain Management Unit, Department of Anaesthesia and Operating Theatre Services, Tuen Mun Hospital, Hong Kong, China; 11Department of Anesthesiology, LKS Faculty of Medicine, The University of Hong Kong, Hong Kong, China; 12Department of Anesthesia and Intensive Care, Faculty of Medicine, The Chinese University of Hong Kong, Hong Kong, China; 13Frankston Pain Management, Frankston, VIC 3199, Australia; 14Department of Anesthesiology, School of Medicine, Chung Shan Medical University, Taichung 40201, Taiwan; 15Department of Anesthesiology, Chung Shan Medical University Hospital, Taichung 40201, Taiwan; 16Department of Anesthesiology, Wan Fang Hospital, Taipei Medical University, Taipei 11031, Taiwan; 17Department of Anesthesiology, School of Medicine, College of Medicine, Taipei Medical University, Taipei 11031, Taiwan; 18Pain Research Center, Wan Fang Hospital, Taipei Medical University, Taipei 11031, Taiwan; 19Department of Anesthesiology, School of Medicine, National Defense Medical Center, Taipei 11490, Taiwan; 20Department of Physical Medicine and Rehabilitation, Tri-Service General Hospital, School of Medicine, National Defense Medical Center, Taipei 11490, Taiwan; crwu98@gmail.com; 21Department of Research and Development, School of Medicine, National Defense Medical Center, Taipei 11490, Taiwan; 22Integrated Pain Management Center, Tri-Service General Hospital, School of Medicine, National Defense Medical Center, Taipei 11490, Taiwan

**Keywords:** neck pain, cervicodiscogenic pain, ultrasound-guided cervical intervertebral disc injection, platelet-rich plasma, motor-sparing analgesia, local anesthesia, biplanar needle validation, ultrasound imaging

## Abstract

Ultrasound-guided needle placement into the cervical intervertebral discs using a lateral-to-medial approach is reportedly possible. Clinically, however, patients commonly present with very high uncovertebral joints or narrowed intervertebral spaces, making the method difficult or impossible. This report presents a novel ultrasound-guided needle placement technique to the cervical intervertebral discs using a more medial approach between the trachea/thyroid gland and the carotid sheath. A patient presented with neck pain radiating to the right shoulder and right-sided interscapular regions that affected his sleep and daily functioning. Physiotherapy, selective nerve root block, and percutaneous endoscopic right C7 laminotomy did not sufficiently improve his condition, which progressed to bilateral interscapular and bilateral shoulder pain. Provocative discography was performed with injection of leukocyte-poor and red blood cell-poor platelet-rich plasma to provoke the discogenic pain, which was treated with platelet-rich plasma mixed with lidocaine. The patient recovered well. A month later, there was a significant decrease in the neck disability index score from the initial 28/50 to 14, and there was a further decrease to 5 after 2 months. In conclusion, this medial approach of ultrasound-guided cervical disc needle placement is feasible, even in patients where disc access by previously described approaches is impossible.

## 1. Introduction

In 2020, Lam et al. described the ultrasound (US)-guided intradiscal injection of platelet-rich plasma (PRP) to the cervical spine to treat cervicodiscogenic pain using a lateral approach (needle trajectory: lateral-to-medial) [1]. The needle passed through the space between the internal jugular vein, carotid artery, and nerve roots and entered the disc covered by the longus coli muscle. However, clinicians commonly encounter patients with severely degenerated discs with annular fibrosus attachments to the vertebral bodies forming large anterolateral osteophytes and almost-fused anterolateral parts of the intervertebral disc and/or patients with tall uncovertebral joints. These conditions usually obstruct the needle from passing the lateral to the medial side underneath the internal jugular veins and carotid artery, entering the disc’s annulus fibrosus. Therefore, reaching the nucleus pulposus of the disc is extremely difficult or impossible when using the aforementioned lateral-to-medial approach. In general, the more lateral the entry, the more difficult it is to insert the needle into the center of the disc.

Therefore, we presented a case of intervertebral disc disease of the cervical spine in which a novel biplanar US-guided intradiscal needle placement was performed using a more medial approach. The trajectory of the needle in this approach corresponds very well to the palpatory method that was first proposed by Cloward in 1959 [2,3].

## 2. Case Report

The patient was a 52-year-old human resources manager who used a smartphone and computer for more than 12 h per day for work. He presented with neck pain that radiated to the right shoulder and right-sided interscapular pain. The pain was more severe at night, affecting his sleep and daily functioning. Ten rounds of a 3-month course of physiotherapy and oral medication, including pregabalin, amitriptyline, nonsteroidal anti-inflammatory drugs, and short courses of oral steroid, failed to improve the symptoms, and his condition progressed to right triceps atrophy. Magnetic resonance imaging (MRI) of the cervical spine confirmed C4/5, C5/6, and C6/7 cervical disc desiccation, with central posterior prolapse that was most severe over the C6/7 level, and compression on the exiting right C7 nerve root.

He was referred to a neurosurgeon, and a diagnostic selective nerve root block (SNRB) was performed, but it did not provide long-lasting relief for more than 2 weeks. Therefore, percutaneous endoscopic right C7 laminotomy was performed 3 months after the SNRB, followed by another 9 months of rehabilitation therapy, which improved his C7 motor power, but the right-sided interscapular pain and right shoulder pain did not respond well. Instead, he developed bilateral interscapular and bilateral shoulder pain, which was more severe at night, disturbing his sleep and daily work.

He was then referred to our musculoskeletal (MSK) center for further treatment. He reported a numerical rating scale (NRS) pain score of 7–8/10 in the daytime and 8–9/10 at night. The neck disability index (NDI) [4] was 28/50, indicating severe disability due to neck pain. MSK examinations showed decreased cervical lordosis and a significantly decreased range of motion in flexion (0–30 degrees), extension (0–30 degrees), side bending (0–15 degrees to the right and 0–30 degrees to the left), and rotation (0–15 degrees to the right and 0–30 degrees to the left). There was local tenderness over the right paraspinal cervical, longus coli, and longus capitis muscles. The physical examination of the right shoulder was essentially normal. Neurological examination reviewed slightly diminished right C7 power, while tendon reflexes and pinprick sensations were normal at all cervical levels.

The nine-month post-laminotomy MRI showed that the nerve root compression over the right C7 existing nerve root had improved markedly. However, the C4/5, C5/6, and C6/7 discs remained desiccated, with similar central broad-based posterior protrusion. The patient desired nonsurgical treatment and opted for US-guided intradiscal PRP injection. Because patient has very high uncovertebral joints and large anterolateral osteophytes formed by calcified annular fibrosus, making the lateral-to-medial approach described before [1] extremely difficult to reach the nucleus pulposus of his discs. A modified medial approach was therefore proposed to the patient, which is explained as follows.

The procedure consisted of two parts. First, we performed a US-guided provocative PRP injection, which involved injecting 0.2–0.4 mL of autologous, leukocyte-free, red blood cell-poor PRP, without local anesthetic, into each disc showing desiccation on MRI and having positive ultrasound-guided disc palpation as explained below, to determine whether this pressure effect could reproduce the patient’s concordant pain, i.e., pain closely resembling his symptomatic pain in intensity and location. Then, if the discogenic pain was reproduced in any one disc, the patient would be injected with another 0.2–0.3 mL of autologous PRP along with 0.1 mL of 2% lidocaine to release the discogenic pain immediately.

Our study was performed following the principles of the Declaration of Helsinki. Written informed consent was obtained from the patient for the use and publication of the case details and related images. Approval by the institutional review board was waived as no identifiable information appears in this case report.

## 3. Technical Note

The patient was administered ceftazidime 1 g (40 mg/mL 25 mL) intravenously 30 min before the procedure. He was placed supine with a towel rolled underneath the neck and upper back to hyperextend the neck and open the anterior part of the cervical spine to increase the space leading to the intervertebral disc as much as possible. The doctor stood alongside the patient where the disc was to be injected. A linear transducer with a broad frequency range (GE L3-12 D, General Electric, Boston, MA, USA) was used, which provided sufficient penetration and resolution. The level of the vertebral body was located by counting upwards or downwards from C6, which has the most prominent anterior transverse process tubercle.

The transverse process of C7 usually has a rudimentary anterior tubercle or only a posterior tubercle. Direct digital palpation of the desiccated discs was performed with ultrasound guidance [5]. The transducer was placed in the sagittal plane of the cervical spine between the carotid artery and the thyroid. The pressure of the transducer usually creates a fascial space and separates the carotid artery from the thyroid, so the discs are underneath the sternocleidomastoid, omohyoid, longus capitis, and longus coli muscles. The center of the transducer was then placed at the disc level to be palpated. Digital pressure was applied through the longus coli muscles (with the longus capitis muscle if above C6) to palpate the anterolateral part of the annulus fibrosus (Figure 1 and Appendix A). The levels of the desiccated discs with tenderness were injected to see if all the concordant pain was reproduced. The other levels, which were not tender on digital palpation, would not be injected, even though MRI showed desiccation.

Dual imaging allowed both real-time B-mode images and the power-Doppler view to visualize the blood vessels. The transducer was first placed in the transverse plane of the neck; an out-of-plane technique was first used. The needle trajectory was to pass between the carotid artery and the thyroid gland. For patients requiring injection above the C4 level, the needle would pass between the carotid artery and the thyroid cartilage. A double-needle technique was used, with a 20-gauge 1-inch hypodermic needle used to enter the skin to pass through the platysma and the sternocleidomastoid (occasionally the omohyoid muscle). Hydrodissection was continued deeper to separate the fascia between the carotid sheath, inferior thyroid artery, and the thyroid gland to reach the longus coli muscle. Then, another 25-gauge 2.5–3-inch spinal needle was used inside the lumen of the 20-gauge needle to continue hydrodissecting through the prevertebral fascia and the sympathetic trunk [6]. The needle passed through the longus coli to reach the anterior surface of the annular fibrosus.

At that time, the transducer was turned 90 degrees to be in the sagittal plane of the cervical spine so that the needle placed over the anterior surface of the cervical spine could be finely adjusted using a biplanar approach to enter the intervertebral space. The needle was then guided to enter the annulus fibrosis for another 1 cm to enter the nucleus pulposus.

The transducer was turned transverse to the cervical spine again to ensure that the needle tip was at the center of the disc, into which 0.2–0.4 mL of autologous leukocyte-free and red blood cell-poor PRP was injected, without local anesthetic, as provocation (Figure 2 and Figure 3, and Appendix A). The patient’s pain level and location were charted. If pain with an NRS score of 8/10 and a similar location was reproduced, another 0.2–0.3 mL of autologous PRP with 0.1 mL of 2% lidocaine was injected to immediately relieve the discogenic pain.

## 4. Results

The US-guided digital palpation elicited tenderness over the C5/6 and C6/7 discs, but the C4/5 disc was not tender. Therefore, provocative PRP injections were performed in both the C5/6 and C6/7 discs. The provocative PRP injection into the C5/6 and C6/7 discs reproduced part of the concordant pain and accounted for the location and severity of the pain he experienced at night. Specifically, the first injection into the C5/6 disc mainly reproduced the bilateral interscapular pain and some right shoulder pain. However, injection into the C6/7 disc mainly reproduced the right shoulder pain with part of the right-sided interscapular pain.

The patient demonstrated good recovery from bilateral interscapular pain and right shoulder pain and there was no notable complication after the procedure. One month after the procedure, he had a 50% improvement in pain, and 2 months after the injection, he had an almost 90% improvement in pain and a markedly increased range of movement of the neck. A month after the procedure, there was a significant decrease in the neck disability index score to 14 and a further decrease to 5 after 2 months. At 9 months postprocedure follow-up, his shoulder or interscapular pain had resolved and his right C7 power normalized. The NRS was 0-1/10, and the NDI was only 2/50. He had occasional neck tightness, which was manageable with self-stretching. He was advised to keep good posture during work and daily activities, do regular stretching exercises, and he was advised to follow up on an as-required basis.

## 5. Discussion

Autologous platelet-rich plasma injections have been shown to be successful in treating 87% of patients with spinal disc herniation and the symptomatic improvement had durable effects up to 8 years postoperatively [7]. In a follow-up assessment of patients who received intradiscal platelet-rich plasma (PRP) injections in a randomized controlled trial for moderate-to-severe lumbar discogenic pain, clinically significant improvements in pain and function at 5–9 years postinjection were demonstrated [8].

The approach presented in this manuscript has all the benefits of the US-guided needle placement in cervical intervertebral discs: there is no radiation hazard to the doctor or patient, and it allows real-time, continuous visualization of the needle and the important soft tissues to prevent damaging those tissues, particularly the carotid artery, the inferior thyroid artery, and the thyroid gland. It also provides a much better angle to access the center of the disc than the pathway illustrated by Lam et al. in their manuscript [1]. This pathway is also particularly suitable for difficult clinical scenarios, such as those involving vastly degenerated discs with severely degenerated annulus fibrosus attachments to the vertebral bodies and the formations of large anterolateral osteophytes, leading to almost complete fusion of the anterolateral parts of the disc, and/or tall uncovertebral joints. This approach has the essential advantage of the needle remaining away from the nerve root, unlike the trajectory of the needle in the approach of Lam et al. [1]. This approach is performed with the patient in supine position and the head and neck supported by a towel rolled underneath the neck and upper back. Apart from opening the anterior neck as much as possible for easier needle entry, the benefit of this setup is that it allows hydrodissection between the carotid sheath, the inferior thyroid artery, and the thyroid gland to reach the intervertebral discs.

Alternatively, an in-plane technique can be used, with the needle in the same trajectory, but with the transducer placed more medially at the neck first in order to obtain a transverse view. After the 20-gauge needle has been placed between the thyroid gland and the carotid artery, the transducer can be turned 90 degrees to validate the direction of the needle using a biplanar approach to let the 25-gauge spinal needle enter the disc in the intervertebral space. After the needle enters the intervertebral space, the transducer is turned to obtain a transverse view of the cervical spine to ensure that the needle can enter the nucleus pulposus of the disc located approximately 1 cm from the rim of the annulus fibrosus (Figure 4 and Figure 5, and Appendix A).

The difficulty in using this alternative approach is that the medial placement of the transducer presses on the cricoid or thyroid cartilage and can cause the patient to choke. Moreover, the post-acoustic shadow of the cartilage and air may hinder the visualization of the structures underneath. A curved linear transducer can be instead used to address these difficulties (e.g., GE C2-9-D Curved Array Probe, General Electric, Boston, MA, USA). It is even more preferable to use a microconvex transducer (e.g., Fujifilm SonoSite C11x Microconvex Probe; Fujifilm SonoSite Inc, Bothell, WA; or GE C3-10-D Microconvex Probe, General Electric, Boston, MA, USA) because the probe size is optimal in that a gentle pressure is sufficient to open a space between the thyroid gland and carotid artery to expose the longus coli and prevertebral fascia underneath the sternocleidomastoid and omohyoid muscles (Figure 6). This can ensure better visualization of the trajectory of either blocking the sympathetic trunk or entering the cervical disc with the initial in-plane technique. Another benefit of using a guidance with microconvex transducer is that these transducers usually have good penetration and resolution so that the whole disc can be visualized (Appendix A).

Occasionally, in some patients in whom the carotid artery and the internal jugular vein are wide apart inside the carotid sheath, the trajectory of the 20-gauge needle may pass between the carotid artery and the internal jugular vein to reach the prevertebral fascia, and the 25-gauge spinal needle subsequently enters the longus coli. This approach requires careful attention to the vagus nerve, which usually courses over the lateral part of the carotid artery and between the carotid artery and the internal jugular vein (Figure 7).

Additionally, even in Lam et al.’s lateral-to-medial method [1], when the needle is placed closer to the lateral side of the internal jugular vein and carotid artery, we recommend using the double-needle technique [2,3,9,10]; a 20-gauge 1.5-inch hypodermic needle should first pass through the skin, platysma, sternocleidomastoid, and anterior scalene until it reaches the prevertebral fascia. Thereafter, another 25-gauge, 2.5-inch spinal needle should pass through the longus coli to enter the disc (Figure 8). This helps prevent severe complications, especially discitis [9]. Moreover, with the double-needle technique, the needle that punctures the disc is much smaller, which can prevent excessive damage. The guiding needle can also help to prevent excessive bending of the thin needle, and guide the fine needle to the proper/correct location.

The US-guided intradiscal technique has some limitations. Because there is no X-ray-guided contrast injection, PRP extravasation to other spaces, such as the epidural spaces, cannot be observed. Nevertheless, as PRP and low-dose lidocaine are used, this approach should have minimal adverse effects. Moreover, it may be difficult to keep the needle in the center of the cervical discs without X-ray guidance and validation. Nonetheless, the needle placement was validated in Lam et al.’s report [1] and the microconvex transducers usually have good penetration and resolution so that the whole disc can be visualized. Moreover, with further advances in US technology and improvements in penetration and image resolution, needle placement in the disc’s center can be achieved, as the lateral mass of the contralateral side of the disc is usually visible. Furthermore, with the technology known as “superb microvascular imaging”, the spread of the injectate inside the desiccated disc can be clearly visualized [5]. Other pitfalls of this techniques are related to the trajectory of the needle, as the needle has to pass through the fascial space between the thyroid (or thyroid cartilage if above C4 level) and the carotid artery. The needle may damage either the thyroid gland, which is highly vascular, the carotid artery, or the inferior thyroid artery. It is, therefore, highly advisable for the doctors to verify the location of the inferior thyroid artery, to use dual-mode imaging during the procedure (if available), and to consistently apply hydrodissection techniques to push away these important structures and open an anechoic space for the needle to follow [11], as shown in Appendix A.

## 6. Conclusions

US-guided cervical disc needle placement is feasible. Herein, we present a novel US-guided needle placement technique for patients with pathological or anatomical difficulties in whom it is impossible to enter the disc using the lateral-to-medial approach. Our new technique uses a medial approach (with biplanar validation). The Section 3 of the manuscript also highlighted some benefits and variations of this medial approach.

## Figures and Tables

**Figure 1 healthcare-10-01427-f001:**
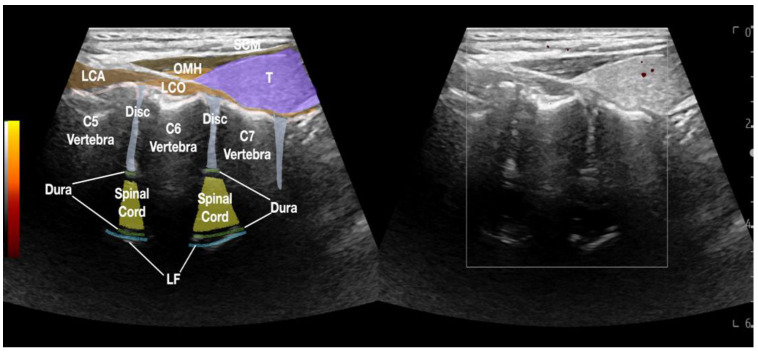
The figure depicts the sonoanatomy of the anterior cervical spine with the transducer in the sagittal plane of the cervical spine in the fascial plane between the carotid artery and the thyroid. Osteophytes were formed at the annulus fibrosus of the C5/6 and C6/7 discs. Visible parts of the anterior and posterior dura and spinal cord are also labeled. Abbreviations: LCA—longus capitis muscle; LCO—longus coli muscle; LF—ligamentum flavum; OMH—omohyoid muscle; SCM—sternocleidomastoid muscle; Spinal Cord—visible parts of spinal cord; T—thyroid gland.

**Figure 2 healthcare-10-01427-f002:**
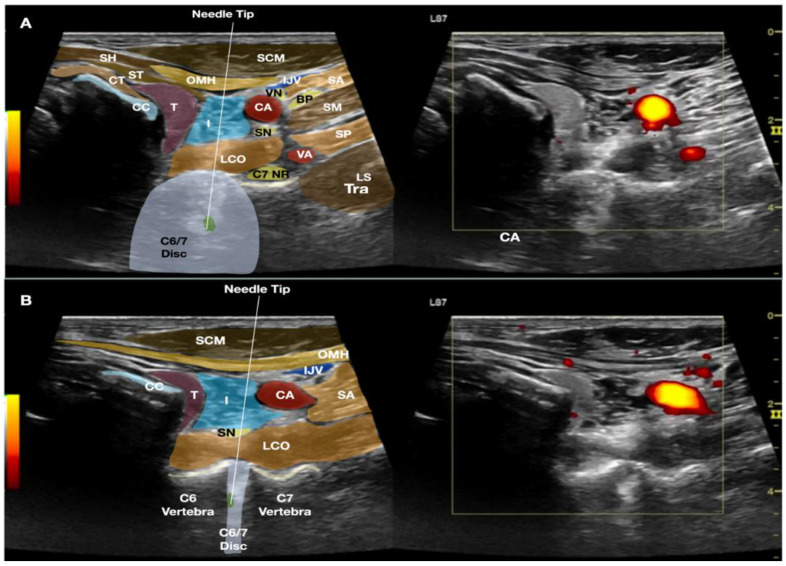
The figure shows the biplanar approach of ultrasound-guided needle placement into the C6/7 intervertebral disc with the trajectory of the needle between the thyroid gland and carotid artery. The image in (**A**) shows the transducer in the transverse plane to the neck and over the level of the C6/7 disc. The image in (**B**) shows that, by pivoting the caudal end of the transducer to the sagittal plane of the neck, the C6 and C7 vertebrae and their intervertebral disc are clearly visible, the needle placement into the nucleus pulposus of the C6/7 disc can be validated. The cricoid cartilage can still be seen in this image due to volume averaging. Abbreviations: BP—brachial plexus (superior trunk); CA—carotid artery; CC—cricoid cartilage; CT—cricothyroid muscle; I—injectate; IJV—internal jugular vein; LCO—longus coli muscle; LS—levator scapulae; NR—nerve root; OMH—omohyoid muscle; SA—scalenus anterior muscle; SCM—sternocleidomastoid muscle; SH—sternohyoid muscle; SM—scalenus medius muscle; SN—sympathetic nerve trunk; ST—sternothyroid muscle; T—thyroid gland; VN—vagus nerve.

**Figure 3 healthcare-10-01427-f003:**
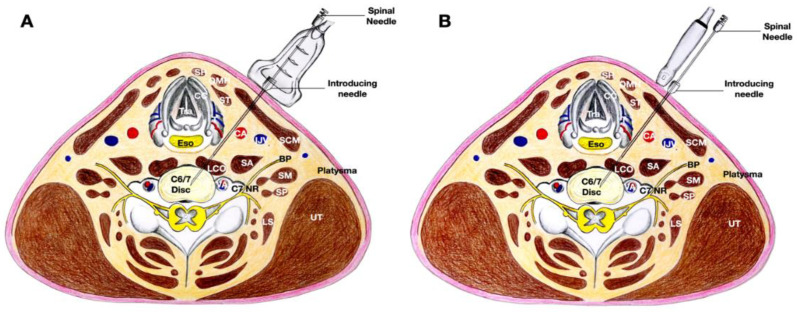
The figure shows the cross-sectional anatomy and the biplanar approach of ultrasound-guided needle placement into the C6/7 intervertebral disc with the trajectory of the needle between the thyroid gland and carotid artery. (**A**) The transducer is in the transverse plane of the neck so an out-of-plane technique is used. (**B**) The transducer is in the sagittal plane of the neck and an out-of-plane technique is again used. Abbreviations: BP—brachial plexus (superior trunk); CA—carotid artery; CC—cricoid cartilage; ESO—esophagus; IJV—internal jugular vein; LCO—longus coli muscle; LS—levator scapulae; NR—nerve root; OMH—omohyoid muscle; SA—scalenus anterior muscle; SCM—sternocleidomastoid muscle; SH—sternohyoid muscle; SM—scalenus medius muscle; ST—sternothyroid muscle; T—thyroid gland; UT—upper trapezius muscle.

**Figure 4 healthcare-10-01427-f004:**
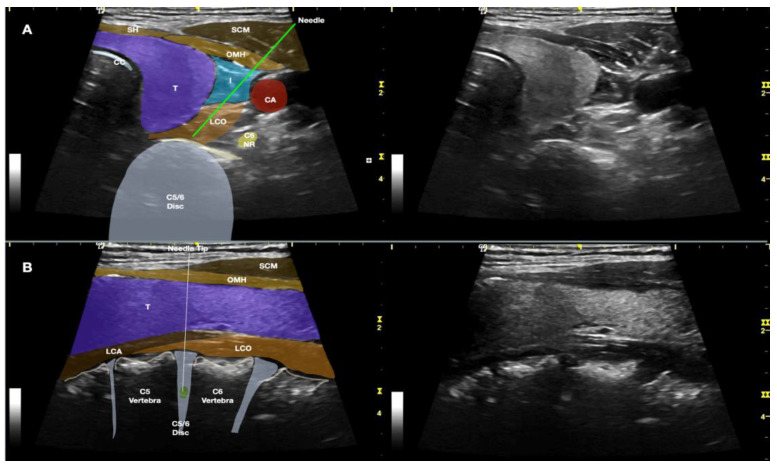
The figure illustrates the same trajectory as that shown in Figure 2, but the procedure was initiated with the transducer placed more medially in order to use the in-plane technique first as shown in (**A**). Upon reaching the annulus fibrosus, the transducer was oriented sagittally to finely adjust the needle to enter the disc as shown in (**B**). Abbreviations: CA—carotid artery; CC—cricoid cartilage; I—injectate; LCA—longus capitis muscle; LCO—longus coli muscle; NR—nerve root, OMH—omohyoid muscle; SCM—sternocleidomastoid muscle; SH—sternohyoid muscle; T—thyroid gland.

**Figure 5 healthcare-10-01427-f005:**
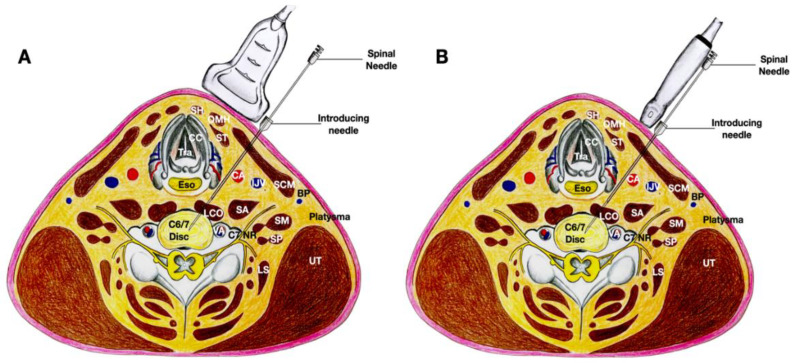
The figure demonstrates the cross-sectional anatomy of the neck at the C6/7 disc, and the needle follows the same trajectory as that shown in Figure 2, but the procedure was initiated with the transducer placed more medially in order to use the in-plane technique first. Upon reaching the annulus fibrosus, the transducer was oriented sagittaly to finely adjust the needle to enter the disc. (**A**) The transducer is in the transverse plane of the neck and placed more medial to the needle. An in-plane needle technique is used. (**B**) The transducer is in the sagittal plane of the neck and the needle is out-of-plane to the transducer. Abbreviations: BP—brachial plexus (superior trunk); CA—carotid artery; CC—cricoid cartilage; ESO—esophagus; IJV—internal jugular vein; LCO—longus coli muscle; LS—levator scapulae; NR—nerve root; OMH—omohyoid muscle; SA—scalenus anterior muscle; SCM—sternocleidomastoid muscle; SH—sternohyoid muscle; SM—scalenus medius muscle; ST—sternothyroid muscle; T—thyroid gland; UT—upper trapezius muscle.

**Figure 6 healthcare-10-01427-f006:**
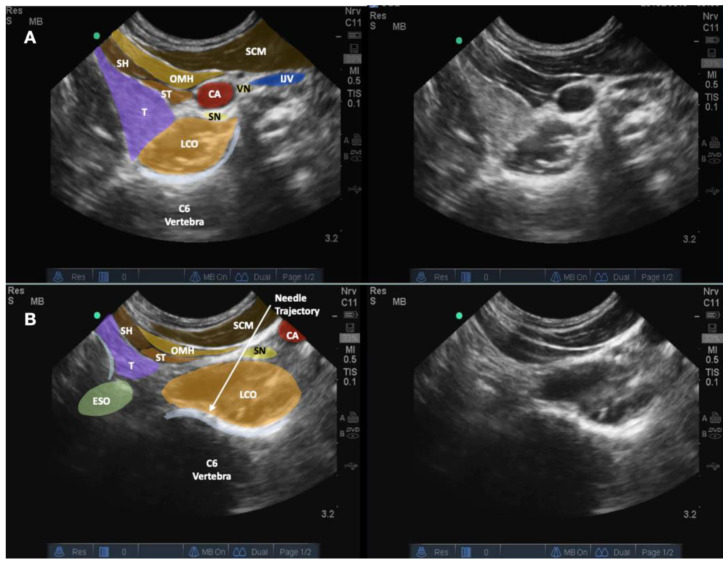
This figure illustrates the benefit of a microconvex transducer. The image in (**A**) displays the sonoanatomy of the anterior cervical structures using a microconvex transducer before gentle transducer compression was applied. The image in (**B**) shows that with application of gentle downward pressure with the transducer; a fascial plane between the carotid artery and thyroid gland is usually created by separating these two important and potentially dangerous soft tissue structures, and the needle trajectory to approach the cervical disc can thus be better visualized and is safer. Abbreviations: CA—carotid artery; ESO—esophagus; IJV—internal jugular vein; LCO—longus coli muscle; OMH—omohyoid muscle; SCM—sternocleidomastoid muscle; SH—sternohyoid muscle; SN—sympathetic nerve trunk; ST—sternothyroid muscle; T—thyroid gland; VN—vagus nerve.

**Figure 7 healthcare-10-01427-f007:**
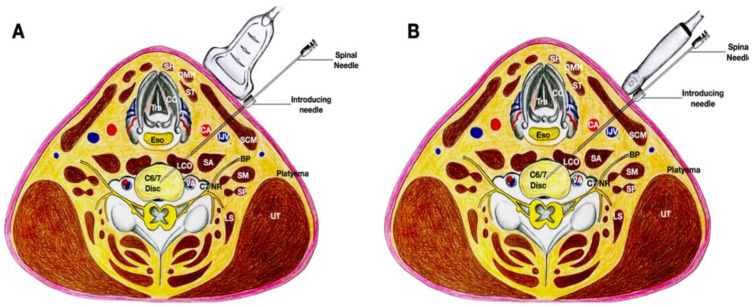
The figure demonstrates a variation of the medial approach, with the needle’s trajectory passing through the space between the carotid artery and the internal jugular vein. The biplanar approach can assist and validate the needle placement. (**A**) The transducer is in the transverse plane of the neck and placed more medially to the needle; an in-plane needle technique is used. (**B**) The transducer is in the sagittal plane of the neck and the needle is out-of-plane to the transducer. Abbreviations: BP—brachial plexus (superior trunk); CA—carotid artery; CC—cricoid cartilage; ESO—esophagus; IJV—internal jugular vein; LCO—longus coli muscle; LS—levator scapulae; NR—nerve root; OMH—omohyoid muscle; SA—scalenus anterior muscle; SCM—sternocleidomastoid muscle; SH—sternohyoid muscle; SM—scalenus medius muscle; ST—sternothyroid muscle; T—thyroid gland; UT—upper trapezius muscle.

**Figure 8 healthcare-10-01427-f008:**
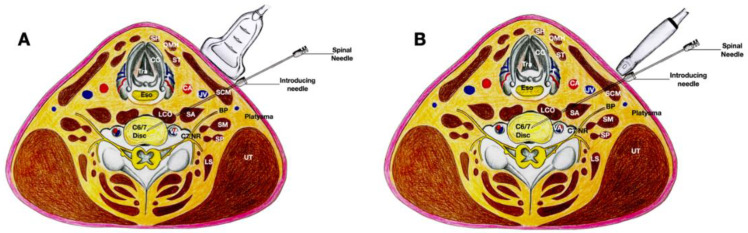
The figure shows Lam et al.’s lateral method [1], with the needle entering closer to the lateral side of the internal jugular vein and passing deeper to the internal jugular vein and carotid artery, using the double-needle technique. (**A**) The transducer is in the transverse plane of the neck and is placed more medially to the needle; an in-plane needle technique is used. (**B**) The transducer is in the sagittal plane of the neck and the needle is out-of-plane to the transducer. Abbreviations: BP—brachial plexus (superior trunk); CA—carotid artery; CC—cricoid cartilage; ESO—esophagus; IJV—internal jugular vein; LCO—longus coli muscle; LS—levator scapulae; NR—nerve root; OMH—omohyoid muscle; SA—scalenus anterior muscle; SCM—sternocleidomastoid muscle; SH—sternohyoid muscle; SM—scalenus medius muscle; ST—sternothyroid muscle; T—thyroid gland; UT—upper trapezius muscle.

## Data Availability

The data that supports the findings of this study are included in this case report.

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
