# Peer review of "Novel Ultrasound-Guided Cervical Intervertebral Disc Injection of Platelet-Rich Plasma for Cervicodiscogenic Pain: A Case Report and Technical Note"

_healthcare, 2022, doi:10.3390/healthcare10081427_

Round 1

Reviewer 1 Report

Comments to the Author
I enjoyed this read and was pleased you present a novel ultrasound-guided needle placement technique such as hydrodissection between the carotid sheath, inferior thyroid artery and the thyroid gland which would make easier and safer than conventional method.  The figures are impressive. However, still needed minor revisions.

1) I would like you to confirm the figure 2 (B) is a correct picture. I think the image in (B) also shows the transducer in the transverse plane like (A) not in the sagittal plane.

2) narrowed intervertebral space(l-38); I don't see how a gap that cannot be pierced from the outside can be easily pierced from the inside. You should remove this, or rephrase so that the meaning goes like this `The more lateral the entry, the more difficult it is to insert the needle into the center of the disc`.

3) The double-needle technique was used because a thin needle is needed to penetrate a narrow gap, but without rigidity, it cannot be guided to the proper location. Did you use the double-needle technique not only for the prevention of infection, but also for this purpose?

4) l would also like you to discuss the advantages of taking your procedure performed in the supine position (hydrodissection between the carotid sheath, inferior thyroid artery and the thyroid gland) Because the stab position can also be made medial without hydrodissection by positioning the body in latera decubitus position (5).

5) Lam et al.` report (l-324); you should show the citation.

6) l would also like you to consider adding more text about technique pitfalls that would occur during the procedure.

7) l-335-336 is not a conclusion drawn from the results presented here.

Reviewer 2 Report

The authors reported a novel ultrasound-guided cervical intervertebral disc injection of platelet-rich plasma for cervicodiscogenic pain. 

Specific comments:

Abstract - No comment

Keywords - Add Ultrasound imaging as a keyword

Introduction - This section is short but this article is a case report with a technical note.

Cas report - For my view, it is not clear if the authors received an approval from ethical committee.

Technical note

The authors have to detail the procedure. For example, the injection volume of ceftazidime; the name of US Scanner...

Results - This section is short but the results are clearly described. I suggest to merge Results and Discussion sections.

Discussion - The results are well interpreted and discussed. Difficulties and limitations of this new procedure are clearly defined.

Conclusion - No comment

Author Response

This manuscript is a resubmission of an earlier submission. The following is a list of the peer review reports and author responses from that submission.